# Efficient Performance of the Methane-Carbon Dioxide Reform Process in a Fluidized Bed Reactor

**José A. Pacífico, Nelson M. Lima Filho and Cesar A. Moraes de Abreu ***

Department of Chemical Engineering, Federal University of Pernambuco, Recife 54521-030, PE, Brazil
* Correspondence: moraesdeabreu@gmail.com; Tel.: +55-81-21268238

**Abstract:** The reforming of methane with $CO_2$ was carried out efficiently in a fluidized bed reactor at 973 K under atmospheric pressure, taking advantage of the nickel catalyst efficiency achieved with a bed of particulate fines. The fluidization operation was characterized by determining a minimum velocity of $3.11 \times 10^{-3}$ ms$^{-1}$ and higher velocities. The reactor worked with surface speeds of up to $1.84 \times 10^{-2}$ ms$^{-1}$, providing conversions from 45% to 51% and a syngas yield of 97%. The control base of the operation focused on the use of $CO_2$ was established through the reaction steps assumed for the process, including methane cracking, reverse Boudouard reaction, and RWGS (reverse reaction of water gas-shift). The reactor designed to operate in two zones was able to simultaneously process surface reactions and catalyst regeneration using feed with 50% excess $CO_2$ in relation to methane. Predictions indicating the production of syngas of different compositions quantified with the $H_2/CO$ ratio from 2.30 to 0.91 decreasing with space-time were validated with the results available for process design.

**Keywords:** methane-$CO_2$ reform; synthesis gas; fluidized bed reactor; selectivity

## 1. Introduction

Through catalytic reform, the transformation of carbonaceous raw materials into gases (hydrogen, synthesis gas) with high thermal content and with potential for intermediate products is carried out, structuring a technological base for the valorization of natural gas, biogas, and emissions [1]. The use of natural gas based on its high methane content is mainly characterized by the production of synthesis gas with different compositions of hydrogen and carbon monoxide [2,3]. An efficient way to recycle two of the most important greenhouse gases is methane-$CO_2$ reforming (DRM) into synthetic gas (syngas) [1]. The DRM alternative has attracted the attention of the industrial sector considering the additional availability of $CO_2$, the low $H_2/CO$ ratio $\approx$ 1:1 of the syngas ($CO + H_2$) produced in this process, to which we add the intention to conduct its operation in a fluidized bed reactor, given the advantages that its operations offer. Much of the carbon dioxide is dispersed in the atmosphere, which makes industrial use of this source unfeasible. Thus, it is worth mentioning the availability of $CO_2$ from localized sources, given the interest in its use. Emissions from refineries, lime, cement production, and alcoholic fermentation are localized sources that can serve to feed subsequent processes. From petroleum processing, carbon dioxide from the regeneration of catalysts by oxidation of coke contributes about 30% of the total emissions of this gas in a refinery, while the fermentation of sucrose from sugarcane, serving the production of alcohol, provides approximately half of the sucrose mass fed in the form of $CO_2$. From methane to hydrogen by catalytic cracking, an important form of carbon appears in the presence of $CO_2$, in addition to preventing the deactivation of the catalytic system, promoting a significant contribution of CO in the composition of the syngas. These two reaction steps characterize dry methane reforming (DRM) as producing syngas [2]. Additions of the WGS reaction with the presence of water lead to the adjustment of the hydrogen content in the syngas composition, approaching

molar equity [3]. In the processing conducted in a fixed bed reactor [4], the limiting event is the carbon deposition on the catalyst by the methane cracking step, which promotes the deactivation and obstruction of the bed. New improved fluidized bed reactors may compensate for these disadvantages [5] Transition metals are active in reforming processes. Among the mentioned elements, Ni is the most used, while Ru and Rh are the most active metals, followed by Ir, Pt, Pd, and Co [6–8]. However, among these, the lower price is an incentive for the use of Ni and Co-based catalysts. In addition to defining the characteristics of the catalyst, in DRM, it is also important to meet the demands arising from industrial production and process intensification [9]. Therefore, the use of structured catalysts can offer several advantages, such as effective mass and heat transfer, lower pressure drop, and operational stability in the chemical regime. The different DRM operations in fixed and fluidized beds, directly assisted by thermal energy and/or solar and plasma energy, have been carried out to guarantee the thermal effects for the process efficiency established by the catalyst activity [10,11]. Reinforcing the resistance of the catalysts to deactivation by carbon deposition and/or sintering, it was decided to operate in a fluidized bed reactor, insisting on the use of a classic nickel-based reforming system. In this case, the structure of the metallic, active phase in fine particles placed as a fluidized bed means a good thermal distribution, avoiding sintering and allowing the catalyst to function in a chemical regime.

In the present work, the results of evaluations of the reforming of methane with an excess of $CO_2$ are presented from operations performed using a structured fluidized bed reactor with a dispersed active nickel catalyst formed by fine non-porous particles. Predictions based on a validated model against operations conducted in the pilot unit of the reactor form the basis for the scheduling of the process.

## 2. Experimental

### 2.1. Catalyst Preparation

Ni (5% wt.)/$\gamma$-$Al_2O_3$ catalysts were prepared by the wet impregnation method. The nickel catalyst containing 5% weight was prepared from a 0.5 molar solution of the precursor salt of $Ni(NO_3)_2 6H_2O$ (99.99%, Sigma-Aldrich, Missouri, USA ) on the catalytic support of pure gamma alumina (Sasol/Catapal). The support samples were initially heat treated at 900 °C for 60 min in a constant argon atmosphere before being used as catalytic support. The prepared catalyst samples were characterized by AAS, surface area measurement by the BET method, X-ray diffraction (XRD), and elementary superficial compositions by XPS.

### 2.2. Processing Evaluation

The methane-carbon dioxide reforming processing experiments were carried out with the nickel catalyst in a fluidized bed reactor under atmospheric pressure. The reactants were fed into a reactor with a gaseous mixture of ($CH_4$:$CO_2$:Ar = 10:15:75 *v/v*) in a flow rate range of $0.60 \times 10^{-2}$ m$^3$h$^{-1}$ to $5.88 \times 10^{-2}$ m$^3$h$^{-1}$, corresponding to GHSV in the range of $0.13 \times 10^{-3}$ m$^3$k$_{gcat}$$^{-1}$h$^{-1}$ to $0.27 \times 10^{-3}$ m$^3$k$_{gcat}$$^{-1}$h$^{-1}$. A thermal sensor and external heating provided, in the reactor, a controlled temperature of 973, 1023, and 1073 K with an accuracy of $\pm 1$ K. The gas samples of the reaction products were collected online from the effluent flow of the reactor. The residual reagents and the products were analyzed by a gas chromatograph (Saturn 2000, Varian, Sao Paulo, Brazil) equipped with a Carbosphere 20/80 (Alltech, S. Marcos, RS, Brazil) column and a thermal conductivity detector.

### 2.3. Description and Characteristics of the Fluidized Bed Unit

Figure 1 shows the catalytic processing unit of the methane reforming with carbon dioxide constituted at its core by the fluidized bed reactor.

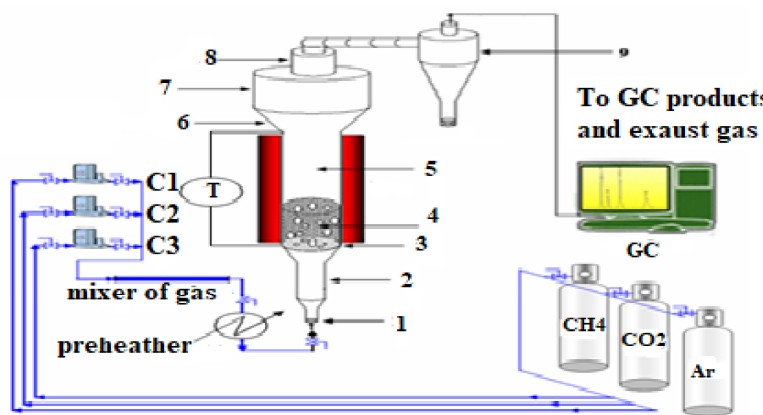

**Figure 1.** Fluidized bed catalytic reforming unit. 1—gas injector, 2—gas disperser, 3—gas distributor plate (porous sintered disc, orifice diameter 55 μm), 4—catalytic bed, 5—reaction zone, 6—transition zone, 7—separation zone, 8—product and exhaust gases outlet exhaust, 9—solids separation cyclone, C1, C2, and C3—gas mass flow controllers, GC—gas chromatography and T—digital pressure loss transducer.

The operating characteristics of the fluidized bed system were as follows: reactor height ($H_t$) 2.24 m, inner diameter ($D_i$) 3.83 × 10$^{-2}$ m, surface operating velocity ($U_o$) 2.41 × 10$^{-2}$ ms$^{-1}$, mean particle diameter ($d_p$) 8.60 × 10$^{-5}$ m, the weight of the catalyst ($m_{cat}$) 280.0 g, particle density ($\rho_p$) 0.75 × 10$^3$ kgm$^{-3}$, and fluid density ($\rho_f$) 1.47 × 10$^{-1}$ kgm$^{-3}$ and fluid viscosity ($\mu_f$) 4.45 × 10$^{-5}$ kg m$^{-1}$s$^{-1}$ were calculated at 973 K and 1.0 bar. The physical behavior of the reactor was evaluated through measurements of the pressure drop in the catalytic bed and according to the surface velocity of the fluid. For this purpose, a bed of alumina supporting the nickel catalyst was used. The operations carried out in the process conditions (973 K, 1.0 bar) allowed us to obtain the pressure drop profiles in the bed.

### 3. Mathematical Modeling

The operational behavior of the reforming process in a fluidized bed reactor is described according to a bubbling bed model, characteristic of fluidization regimes with small particles and high fluid ascent speeds in the bed. The occurrence-based design is used with two distinct phases (diluted and dense) flowing simultaneously in the bed [8,12]. Convective flows with axial dispersions are considered for both phases; mass transfer from the dilute phase to the dense phase and reactive operation in the chemical kinetic regime. The operations take place considering $u_b$ is the bubble velocity, $u_{mf}$ is the minimum fluidization velocity, and $\varepsilon_{mf}$ is the void fraction in incipient fluidization. Isothermal and stationary mass balances were elaborated to describe the process operations in a fluidized bed reactor, referring to the components "i" of the reaction medium involving the reaction steps (j). The bubbling bed model [13] focused on the two phases, bubble and emulsion that are considered the convective and axial dispersion effects, the mass transfer from the bubble phase to the emulsion phase, and the reaction steps. The mass balance equations (Equations (1) and (2)) developed were written as:

- For the bubble phase:

$$D_e \frac{d^2 C_{iB}}{dz^2} - u_z \frac{dC_{iB}}{dz} - K_{BE}(C_{iB} - C_{iE}) + R_{iB} = 0 \tag{1}$$

- For the emulsion phase:

$$(1 - \varepsilon_B)D_e \frac{d^2 C_{iE}}{dz^2} - (1 - \varepsilon_B)u_z \frac{dC_{iE}}{dz} + \varepsilon_B K_{BE}(C_{iB} - C_{iE}) + \\ (1 - \varepsilon_B)\left(1 - \varepsilon_{mf}\right) R_{iE} = 0 \tag{2}$$

where $C_{i_B}$, $C_{i_E}$ are the concentrations of the components at each phase, and consumption and production reaction terms $(R_{iB}\ R_{iE})$ are included as negative and positive, respectively. The corresponding boundary conditions are:

$$z = 0,\ C_{i0B}^{-} = C_{iB}^{+} - \frac{De}{uo}\left[\frac{C_{iB}}{dz}\right]_{z=0} ;z = L,\ \left[\frac{dC_{iB}}{dz}\right]_{z=L} = 0 \qquad (3)$$

$$z = 0,\ C_{iE0} = 0 ; z = L,\ \left[\frac{dC_{iE}}{dz}\right]_{z=L} = 0 \qquad (4)$$

In which, $\varepsilon_B$, $\varepsilon_L$, $1 - \varepsilon_B$ and $1 - \varepsilon_L$ are the bubble fraction in the bed, bed porosity, solids fraction in the bubble phase, and solids fraction in the dense phase, respectively.

The coupled and non-linear second-order ordinary differential equations (ODEs) form a system that resorts to a numerical solution (finite differences), transforming the problem from a continuous domain to a discrete domain with nodal points. The equations become algebraic, and the system is solved using mathematical optimization software (Scilab software, version 6.1.0).

## 4. Results and Discussion

### 4.1. Catalyst Formulation

The nickel content and surface area of the catalyst were 4.82% by weight and $1.56 \times 10^2$ m$^2$g$^{-1}$, respectively. The solid phases of the catalyst identified by XRD were the $\gamma$-Al$_2$O$_3$ support, the nickel metallic phase Ni$^\circ$ ($2\theta$ = 50.2, 84.9 and at 92.8), and NiO ($2\theta$ = 9.8, 18.6, 44.8, 62.4 and 76.5). Additional phases were identified as Ni$_3$C and NiAl$_2$O$_4$. The surface elemental compositions (XPS, %) atomic were 24.00% C, 0.82% Ni, 1.17% Na, 1.42% Si, 28.74% Al, and 43.24% O, indicating the moderate deposition of carbon in the catalyst after the reaction.

### 4.2. Preliminary Physical Evaluations

The evaluation of the fluidization regime was carried out in experiments at 973 K for $\gamma$-Al$_2$O$_3$ particulates (280.0 g, dp 86 μm) operated with a bed of 1.00 m, where pressure drops were measured for velocities surface of up to $1.8 \times 10^{-2}$ m s$^{-1}$, in increasing and then decreasing evolution, which by hysteresis characterizes the operational domain. Figure 2 shows the pressure drop profile.

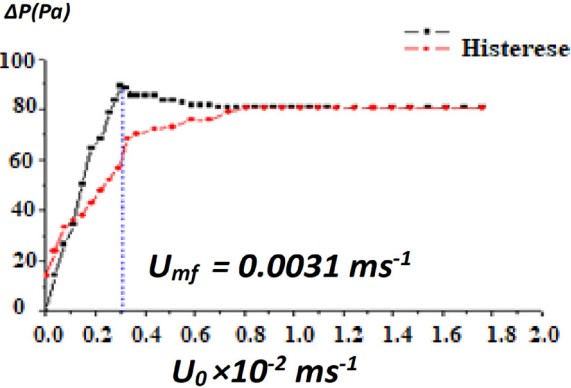

**Figure 2.** Pressure drop profiles as a function of the surface gas velocity. Conditions: 280.0 g, dp 86 μm, $\gamma$-Al$_2$O$_3$, 1.0 bar, 973 K.

It was observed in the profiles that when the superficial velocity was increased, the pressure drop increased until reaching a maximum value of 88 Pa at 0.31 ms$^{-1}$. Such effort indicates that the bed expanded in the gas. Then, continuing to increase the superficial velocity, the pressure drop remained at the 80 Pa level in the evaluated range of up to 1.8 ms$^{-1}$.

The fluidization dynamic parameters bed porosity ($\varepsilon_L$) and bed particle height ($H_{PL}$) were experimentally determined at 973 K. The minimum fluidization velocity ($U_{mf} = 0.31 \times 10^{-2}$ m s$^{-1}$) was also estimated at 973 K according to the evaluation obtained as a function of the pressure loss (Equations (5) and (6)), which was adjusted by the equation of Ergun [12], with an acceptable mean relative deviation of 17.32%.

$$\frac{1.75}{\varepsilon_m^3 \phi}\left(\frac{\rho_f d_p U_{mf}}{\mu}\right)^2 + 150\frac{(1-\varepsilon_m)^2}{\varepsilon_m^3 \phi^2}\left(\frac{\rho_f d_p U_{mf}}{\mu}\right) = \frac{d_p^3 \rho_f (\rho_s - \rho_f)}{\mu^2}g \tag{5}$$

$$\frac{\Delta P}{L} = 150\frac{(1-\varepsilon_m)}{\varepsilon_m^3}\left(\frac{\mu U_0}{[\phi d_p]^3}\right) \tag{6}$$

The velocity $U_{mf}$ was located, according to the representation in Figure 2, in a transition zone between the fixed bed region ([0.0–0.31] $\times 10^{-2}$ m s$^{-1}$) and the fluidized bed region ([0.32–1.8] $\times 10^{-2}$ m s$^{-1}$), following a continuous expansion of the bed.

### 4.3. Process Evaluation

Observing the results of operational evaluations of methane reforming using higher $CO_2$ content ($CH_4$:$CO_2$/2:3 mol), the mechanism and kinetics proposition validated through the work of Abreu et al. (2008) [14] was accepted as plausible to be included in the model that represents DRM operations. The related chemical equations are expressed below in Table 1, along with the corresponding expressions of the reaction rates.

**Table 1.** Mechanism and reaction kinetics.

| Reaction Step | Reaction Rate | |
|---|---|---|
| (I) $CH_4 \rightarrow C + 2H_2$, $k_1$/methane cracking | $R_{CH_4} = \dfrac{k_1 K_{ad} C_{CH_4}}{1 + K_{ad} C_{CH_4}}$ | (7) |
| (II) $CO_2 + H_2 \rightarrow CO + H_2O$, $k_2$/rWGS (III) $C + CO_2 \rightarrow 2CO$, $k_3$/Boudouard reverse | $R_{CO_2} = k_2\left(C_{CO_2}C_{H_2} - \dfrac{1}{K_{eq2}}C_{CO}C_{H_2O}\right) + k_3\left(C_{CO_2} - \dfrac{1}{K_{eq3}}C_{CO}\right)$ | (8) |

The reaction rates $Ri$ are included in the model equations (Equations (1) and (2)), where for the reactants they are $R_{CH_4}$ and $R_{H_2}$ (Equations (8) and (9)) and for the products they are written as $R_{H_2} = 2R_{CH_4} + (R_{CO_2})^{II}$, $R_{CO} = (R_{CO_2})^{II} + 2(R_{CO_2})^{III}$, $R_{H_2O} = (R_{CO_2})^{II}$. The kinetic parameters that quantify the steps of the adopted mechanism have their values expressed below and are valid for the process temperature 753 K, under 1.0 bar (Abreu et al. (2008) [14]):

$k_1 = 8.23 \times 10^{-5}$ mol g$_{cat}^{-1}$s$^{-1}$, $k_2 = 3.38 \times 10^{-7}$ mol g$_{cat}^{-1}$s$^{-1}$, $k_3 = 1.85 \times 10^{-4}$ mol g$_{cat}^{-1}$s$^{-1}$
$K_{ad} = 8.23 \times 10^{-5}$ m$^3$ mol$^{-1}$, $K_{eq2} = 0.62$

At the operating conditions, to confirm the rate-controlling regime related to the catalyst, the mass transfer limitations estimated through the Weisz criterion ($\Phi_j = r_i L_c^2/D_e C_i$, De effective diffusivity), and the external mass transfer resistance fraction $f_{ie}$ ($f_{ie} = r_i L_c/k_{im}C_i$, $k_{im}$ mass transfer coefficient; Villermaux [15]) were quantified for methane and carbon monoxide at 973 K, respectively, as follows: ($\Phi_j = 3.13 \times 10^{-3}$, $1.17 \times 10^{-2}$; $f_{ie} = 3.23 \times 10^{-2}$, $3.37 \times 10^{-2}$. The estimated values ($\Phi_j \rightarrow 0$; $f_{ie} < 0.05$) show that the catalytic process is rate-controlling. The estimated values, located in the domains of $\Phi_j \rightarrow 0$ and $f_{ie} < 0.05$), show that the reaction rate controls the catalytic process.

The interphase mass transfer of the components included in the model (Equations (1) and (2)) is quantified specifically by the global bubble-emulsion mass transfer coefficient

($K_{BE}$) through the correlation given by Equation (9) [16], where $D_{AB}$ is the molecular diffusivity (A: $CH_4$, $CO_2$/B: Ar) and $d_b$ is the mean bubble diameter.

$$K_{BE} = 4.5\left(\frac{U_{mf}}{d_b}\right) + 5.85\frac{D_{AB}^{1/5}g^{1/4}}{d_b^{5/4}} \tag{9}$$

### 4.4. Experimental Evaluation and Validation

Under reactive fluidization conditions, 280.0 g of Ni catalyst (4.86% wt./$\gamma$-$Al_2O_3$), previously formulated and characterized, was used in the reactor with volumetric flow ranging from $1.0 \times 10^{-5}$ m$^3$ h$^{-1}$ to $2.0 \times 10^{-5}$ m$^3$ h$^{-1}$, and an $U_o/U_{mf}$ ratio 5.68. The evolution of the concentration fractions (<$C_i$>/$C_{i0}$) of the reactants ($CH_4$, $CO_2$) and products (CO, $H_2$, $H_2O$) and effluents observed and predicted by the model are shown in Figures 3 and 4.

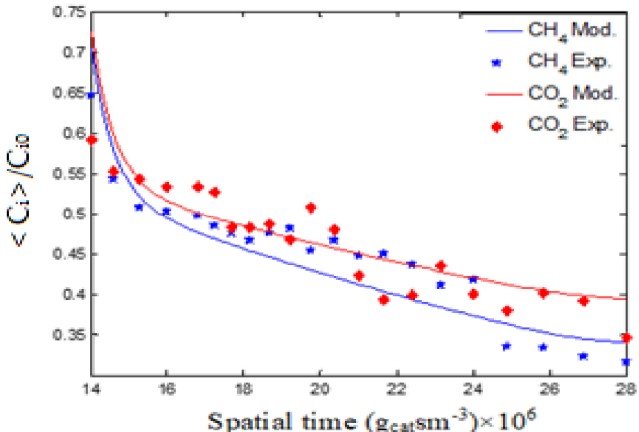

**Figure 3.** Evolutions of reagent concentrations as a function of spatial time. Conditions: Ni (4.86% wt.)/$\gamma$-$Al_2O_3$ catalyst, fluidized bed (280.0 g), 1.0 bar, 973K.

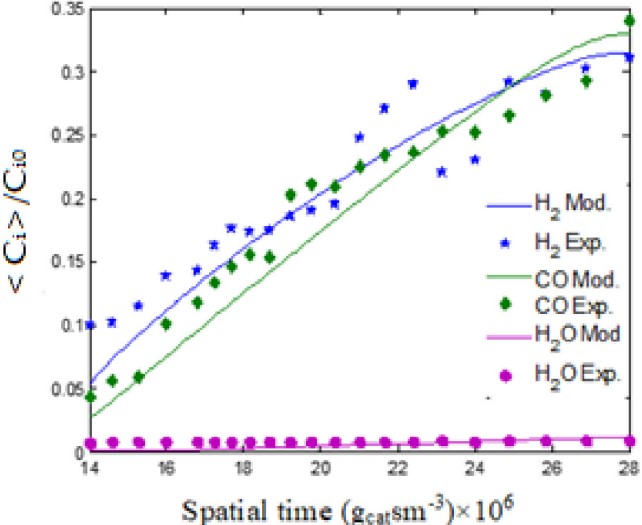

**Figure 4.** Evolutions of product concentrations as a function of space-time. Conditions: Ni (4.86% wt.)/$\gamma$-$Al_2O_3$ catalyst, fluidized bed (280.0 g), 1.0 bar, 973K.

In the domain expressed by GHSV = [1.29 − 2.57]$\times 10^{-4}$ m$^3$g$_{cat}^{-1}$h$^{-1}$ and molar feed composition of $CH_4$:$CO_2$:Ar = 2:3:15, the conversions of reactants at the reactor outlet ($X_i = 10^2$ [$C_{i0} − C_j$]$C_{j0}^{-1}$; i = $CH_4$, $CO_2$) are represented in Figure 5, where the curves reached levels of 51.54% and 45.51% for methane and carbon dioxide, respectively.

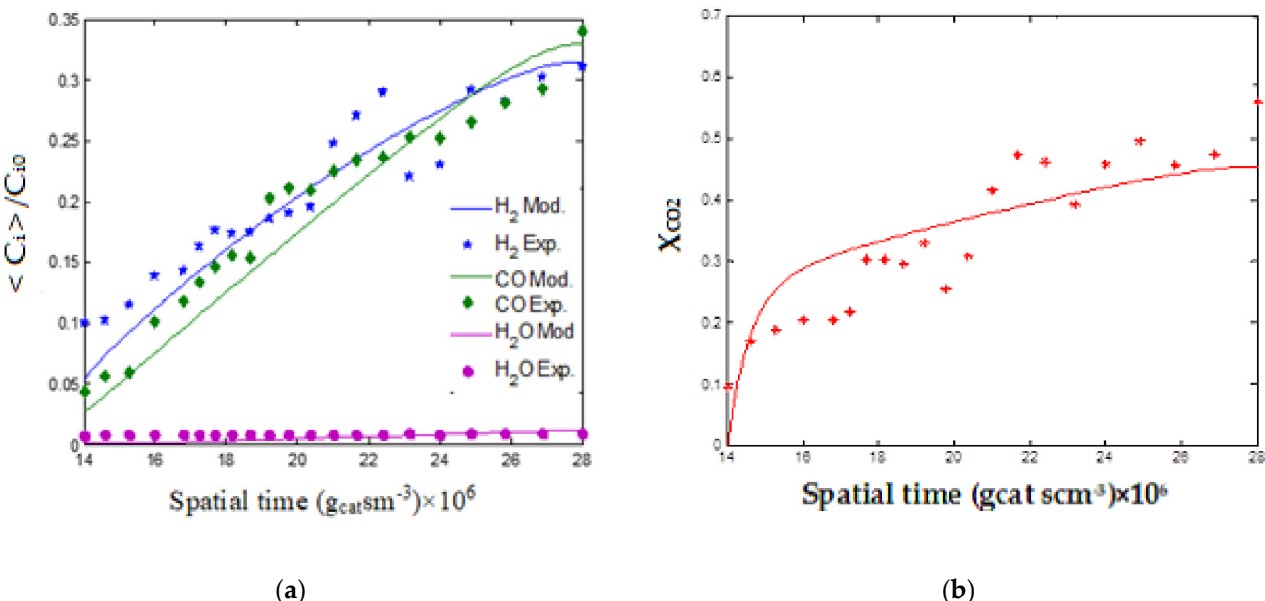

(**a**)           (**b**)

**Figure 5.** Evolutions of conversions as a function of space-time. (**a**) $CH_4$ (**b**) $CO_2$. Conditions: Ni (4.86 % wt.)/$\gamma$-$Al_2O_3$ catalyst, fluidized bed (280.0 g), 1.0 bar, 973K.

The evolution of the yields ($Y_i = 10^2 C_i (C_{CH_40} + C_{CO_20})^{-1}$; $i = H_2$, $CO$) of hydrogen and carbon monoxide reached 47.72% and 49.95%. In Figure 6, these yields are represented as evolution in terms of synthesis gas yield ($Y_{syngas} = 10^2 [C_{H_2} + C_{CO}](C_{CH_40} + C_{CO_20})^{-1}$), highlighting an increasing curve that converges to reach 97.67% of yield.

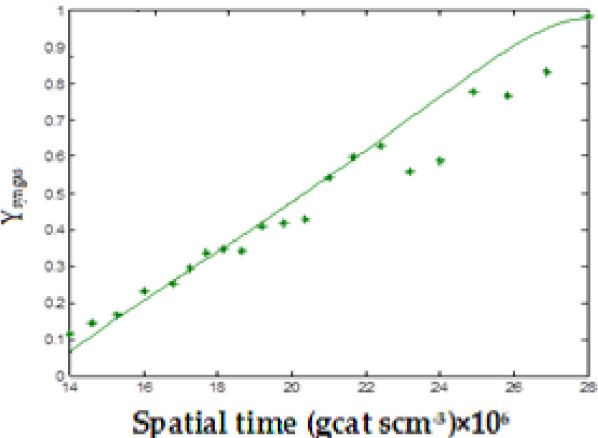

**Figure 6.** Evolution of syngas yield as a function of space-time. Conditions: Ni (4.86 % wt.)/$\gamma$-$Al_2O_3$ catalyst, fluidized bed (280.0 g), 1.0 bar, 973K.

Predictions indicating the production of syngas of different compositions quantified through the $H_2$/CO ratio from 2.3 to 0.91 decreasing with spatial time were validated with the results available for process design.

## 5. Conclusions

The methane-carbon dioxide reforming process was carried out efficiently in a chemical kinetic regime using the system composed of a nickel catalyst formulated as Ni-NiO (4.82% wt.)/$\gamma$-$Al_2O_3$, pulverized (dp 86 μm), forming 280.0 g of a fluidized bed as a continuous reactor (FBR).

The performance of the catalyst, highlighting its activity and stability for synthesis gas production, was achieved with thermal stability and distribution uniformity of the

bubble-emulsion system. A chemical kinetic regime characterized the operation (reduced dp ~2.0 mm) feeding with excess $CO_2$ ($CO_2$:$CH_4$/3:2).

Increasing yields in syngas equimolar are achieved according to the practiced space-time. The control base of the operation focused on the use of $CO_2$ was established through the reaction steps assumed for the process, including methane cracking, reverse Boudouard reaction, and RWGS (reverse reaction of water gas-shift).

In the fluidized bed, it was possible to convert methane into hydrogen and carbon dioxide into CO. Carbon deposition was eliminated, and the consumption of hydrogen by the RWGS was avoided. The reactor designed to operate in two zones was able to simultaneously process surface reactions and catalyst regeneration using feeds with 50% more $CO_2$ than methane.

The hydrogen and carbon monoxide yields, reaching 47.72% and 49.95%, allowed us to obtain 97.67% for the synthesis gas. Steady-state predictions based on a heterogeneous model validated against the operations carried out in the pilot reactor unit form the basis for the process design. From the perspective of using the gaseous product for various syntheses, different compositions quantified based on the $H_2$/CO ratio revealed a variation from 2.3 to 0.91, according to the increase in space times.

**Author Contributions:** Conceptualization and methodology, J.A.P., formal analysis and investigation, N.M.L.F., and revision and editing C.A.M.d.A. All authors have read and agreed to the published version of the manuscript.

**Funding:** This research received no external funding.

**Institutional Review Board Statement:** Not applicable.

**Informed Consent Statement:** Not applicable.

**Acknowledgments:** Acknowledgments from the authors to the Federal University of Pernambuco, Brazil, and to the CNPq (National Council of Science and Technology), Brazil, for their academic and structural support, whose results form the basis of this work.

**Conflicts of Interest:** The authors declare no conflict of interest.

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
