# Peer review of "Efficient Performance of the Methane-Carbon Dioxide Reform Process in a Fluidized Bed Reactor"

_methane, doi:10.3390/methane2010004_

Round 1
Reviewer 1 Report
I have attached file to this email.

Author Response
REVIEWER 1
1.1. There are several types and grammatical errors that often make it difficult to understand the text.
Authors: The text of the manuscript was revised.
1.2.An outline of the article is recommended at the end of the introduction section
Authors: An outline of the article has been added before the last paragraph of the Introduction, pp 63-71.
1.3.The innovation of this work is not highlighted. Clearly explain what the latest the progress and past work of this document is based on?
Authors: The performance of the catalyst, highlighting its activity and stability for the production of a synthesis gas with H2/CO = 1 ratio, was achieved through the use of FBR.
It was demonstrated that under thermal stability and emulsion distribution uniformity, operating in chemical kinetic regime (reduced dp ~ 2.0 mm), and feeding excess CO2 (CO2:CH4/1.5:1), gas yields of Equimolar synthesis are achieved according to the space time practiced.
The highest equimolar synthesis gas yields are close to those predicted by thermodynamic equilibrium.
In previous works (Abreu et al. [11]) we carried out methane reforms in a fixed bed, addressing operations whose results indicated the need to process in a chemical kinetic regime. Thus, we have advanced by adopting operations in a fluidized bed, meeting the control conditions of the surface reaction steps and being able to avoid in situ deactivation by carbon deposition.
1.4.Did the authors compare the results of their study with the work of other authors?
Authors: Fluidized bed reactors are consolidated for use mainly for the maximized production of relatively light products (Steynberg, A. and Dry, [14]), in comparison with the fixed bed, mainly to operate natural gas conversion processes.
1.5. Clearly explain Figure 2.
Authors: Details about Figure 2 were included in the paragraph after its presentation.
1.6. Authors need clarification on how to solve equation 5.
Authors:
The coupled and non-linear second order ordinary differential equations (ODEs) form a system that resorts to a numerical solution (finite differences) transforming the problem from a continuous domain to a discrete domain with nodal points. The equations become algebraic and the system is solved using mathematical optimization software, pp154-158.
1.7. The conclusion is brief and does not cover all aspects of the work.
Authors: The Conclusion item was expanded detailing the main aspects of the work.
1.8.Some important findings can be presented in the Abstract.
Authors: Additional findings are included in the Abstract.
1.9.Current references are appropriate. Some new sources on the research topic should be added to the list of sources and mentioned in the text.
Authors: References are added and mentioned
Reviewer 2 Report
Nice work. Please see your submitted PDF manuscript, with my extensive comments.

Author Response
REVIEWER 2
2.1. Nice work. Please see your submitted PDF manuscript, with my extensive comments.
Authors: More detailed comments are included in the Abstract, Introduction, Conclusions, in Figure 2.

Reviewer 3 Report
I recommend the publication of this paper in Methane after a major revision according to the following comment:
1- What is the novelty of this work? The novelty of the work should be highlighted.
2- It is suggested that the nickel content of the catalyst be measured by inductively coupled plasma (ICP) analysis.
3- Some relevant modern references should be added to the text.
4- A few typing errors need to be corrected. For example: Page 6, line 182: CO2; should be CO2.
Author Response
REVIEWER 3
3.1. It is suggested that the nickel content of the catalyst be measured by inductively coupled plasma (ICP) analysis.
Authors: The catalyst prepared from the precursor salt Ni(NO3)2.6H2O with a predicted metal content of 5% by weight, resulted in the system Ni (4.65% by weight)/γ-Al2O3 according to analysis in atomic absorption spectrometry .
3.2. Some relevant modern references should be added to the text.
Authors: References are added and mentioned.
3.3. A few typing errors need to be corrected. For example: Page 6, line 182: CO2; should be CO2. Authors: Corrections were made.
Round 2
Reviewer 1 Report
The Authors have correctly made all the required changes.
Reviewer 2 Report
Thank you for your attention to my comments.
Reviewer 3 Report
The authors have revised the manuscript follow the comments. I suggest to accept the manuscript in its current form for publication in the Methane. Only, typing errors should be corrected in the revised manuscript. Page 1, line 23, and Page 7, lines 221, 222, and 223: CO2, CH4, and H2 should be CO2, CH4, and H2.